Comparison of the clinical value of MRI and plasma markers for cognitive impairment in patients aged ≥75 years: a retrospective study

Wang Wei 1 2 3 4
Shi Lin 5
Ma Hong 2 3 4
Zhu Shiguang 6
Ge Yaqiong 7
Xu Kai 1 2 3 4 xukai@xzhmu.edu.cn
1 Nanjing Medical University , Nanjing, Jiangsu , China
2 Department of Radiology, The Affiliated Hospital of Xuzhou Medical University , Xuzhou, Jiangsu , China
3 School of Medical Imaging, Xuzhou Medical University , Xuzhou, Jiangsu , China
4 Institute of Medical Imaging and Digital Medicine, Xuzhou Medical University , Xuzhou, Jiangsu , China
5 Department of Radiology, The Second Affiliated Hospital of Shandong First Medical University , Taian, Shangdong , China
6 Department of Neurology, The Affiliated Hospital of Xuzhou Medical University , Xuzhou, Jiangsu , China
7 GE Healthcare, Precision Health Institution , Shanghai , China
Li Tian
Electronic publication date: 2023 Jun 22
Publication date: 2023
Volume: 11
Electronic Location ID: e15581
Received 2023 Feb 21; Accepted 2023 May 26
Copyright: © 2023 Wang et al.
Copyright year: 2023
Copyright holder: Wang et al.
License: This is an open access article distributed under the terms of the Creative Commons Attribution License, which permits unrestricted use, distribution, reproduction and adaptation in any medium and for any purpose provided that it is properly attributed. For attribution, the original author(s), title, publication source (PeerJ) and either DOI or URL of the article must be cited.
License URL: https://creativecommons.org/licenses/by/4.0/

Keywords: White matter, Leukoencephalopathies, Brain atrophy, Amyloid-beta-peptides, Tau proteins, Cognitive impairment

Funding: National Natural Science Foundation of China 81771904 This study was funded by the National Natural Science Foundation of China (Grant No. 81771904). The funders had no role in study design, data collection and analysis, decision to publish, or preparation of the manuscript.

==============================
Background

Dementia has become the main cause of disability in older adults aged ≥75 years. Cerebral small vessel disease (CSVD) is involved in cognitive impairment (CI) and dementia and is a cause of vascular CI (VCI), which is manageable and its onset and progression can be delayed. Simple and effective markers will be beneficial to the early detection and intervention of CI. The aim of this study is to investigate the clinical application value of plasma amyloid β1-42 (Aβ42), phosphorylated tau 181 (p-tau181) and conventional structural magnetic resonance imaging (MRI) parameters for cognitive impairment (CI) in patients aged ≥75 years.

Methods

We retrospectively selected patients who visited the Affiliated Hospital of Xuzhou Medical University and were clinically diagnosed with or without cognitive dysfunction between May 2018 and November 2021. Plasma indicators (Aβ42 and p-tau181) and conventional structural MRI parameters were collected and analyzed. Multivariate logistic regression and receiver operator characteristic (ROC) curve were used to evaluate the diagnostic value.

Results

One hundred and eighty-four subjects were included, including 54 cases in CI group and 130 cases in noncognitive impairment (NCI) groups, respectively. Univariate logistic regression analysis revealed that the percentages of Aβ42+, P-tau 181+, and Aβ42+/P-tau181+ showed no significant difference between the groups of CI and NCI (all P > 0.05). Multivariate logistic regression analysis showed that moderate/severe periventricular WMH (PVWMH) (OR 2.857, (1.365–5.983), P = 0.005), lateral ventricle body index (LVBI) (OR 0.413, (0.243–0.700), P = 0.001), and cortical atrophy (OR 1.304, (1.079−1.575), P = 0.006) were factors associated with CI. The combined model including PVWMH, LVBI, and cortical atrophy to detect CI and NCI showed an area under the ROC curve (AUROC) is 0.782, with the sensitivity and specificity 68.5% and 78.5%, respectively.

Conclusion

For individuals ≥75 years, plasma Aβ42 and P-tau181 might not be associated with cognitive impairment, and MRI parameters, including PVWMH, LVBI and cortical atrophy, are related to CI. The cognitive statuses of people over 75 years old were used as the endpoint event in this study. Therefore, it can be considered that these MRI markers might have more important clinical significance for early assessment and dynamic observation, but more studies are still needed to verify this hypothesis.

Introduction

The incidence of dementia is rising around the world (Jia et al., 2020), which is considered to be one of the global public health events and has become the main cause of disability in older adults aged ≥75 years in China (Jia et al., 2020). Cerebral small vessel disease (CSVD) is involved in the pathophysiology of cognitive impairment (CI) and dementia in older adults. In addition, it is a cause of vascular CI (VCI) (Goda et al., 2021; Kling et al., 2013; Mustaph et al., 2019). VCI can coexist with many neurodegenerative problems, such as Alzheimer’s disease and other systemic conditions, potentially resulting in cognitive impairment (Jellinger, 2007). VCI is considered a modifiable factor with potentially important implications for prevention and treatment of cognitive impairment and dementia (Jellinger, 2007).

White matter hyperintensities (WMH) and brain atrophy are two common imaging markers of CSVD. WMH can be observed in a relatively large proportion of older adults (Wardlaw et al., 2013) and is realted to the loss of cognitive functions and dementia (Wang et al., 2020; Hu et al., 2021). Brain atrophy on imaging is shown as a reduction of brain volume that is not associated with a macroscopic focal injury (Wardlaw et al., 2013). Tissue loss is assumed to arise from enlargement of the peripheral (sulci) and central (ventricle) CSF Spaces (Wardlaw et al., 2013). In healthy individuals, brain atrophy acceleration is observed after 60 years (Hedman et al., 2012). Brain atrophy plays an important role in the pathogenesis of dementia (Al-Otaibi et al., 2020; Jung et al., 2020).

Pathologies of CSVD include arteriolosclerosis and cerebral amyloid angiopathy, involving the deposition of amyloid β (Aβ) (mainly Aβ1-40 (Aβ40) and Aβ1-42 (Aβ42)) in the cerebrovascular system (Viswanathan & Greenberg, 2011). Aβ can lead to hyperphosphorylation of tau protein, resulting in Aβ-induced neuronal toxicity, memory impairment, and learning disabilities (Nussbaum et al., 2012). VCI can be isolated, however, it is frequently combined with Aβ pathology (Leijenaar et al., 2020; Kapasi, DeCarli & Schneider, 2017). However, there is no evidence that CSVD or VCI is related to Aβ42 and P-tau181, and there are different methods (such as PET, CSF and plasma examination) to evaluate Aβ or tau load (Kaffashian et al., 2014; Hilal et al., 2017; Garnier-Crussard et al., 2020; Caballero et al., 2020). In contrast to functional MR sequences, PET or lumbar puncture, conventional structural MRI and plasma examination are simultaneous, cost-effective and noninvasive approaches, which are essential for clinical application and scientific research, especially for large-scale or longitudinal studies.

To date, there are few studies on plasma Aβ and p-tau and their relationship or studies comparing these markers with WMH and brain atrophy on conventional MRI. Here, we aimed to investigate and compare the clinical application value of plasma Aβ42, p-tau181 and conventional structural MRI parameters in assessment CI in patients aged ≥75 years.

Materials and Methods

The retrospective study was conducted in accordance with the principles of the Declaration of Helsinki. The requirement for informed consent was waived, which was approved by the ethics committee of the Affiliated Hospital of Xuzhou Medical University (XYFY2022-KL331-01).

Patient selection

We retrospectively selected all patients who visited the Affiliated Hospital of Xuzhou Medical University and were clinically diagnosed with or without cognitive dysfunction between May 2018 and November 2021.

The inclusion criteria were (1) clinical diagnosis of CI or noncognitive impairment (NCI), (2) aged 75 years and over, (3) available plasma Aβ42 and P-tau181 test results, (4) an MRI examination was performed, and (5) the interval time between the blood test and MRI was <1 week. The exclusion criteria were (1) brain disease, such as massive infarction, hemorrhage, malacosis or tumor, (2) brain injury, (3) toxic or metabolic diseases, or (4) medication history that affects cognitive function or can lead to WMH.

Data collection

The clinical data of the patients were extracted from the patient charts, including age, sex, cognitive status, medical history, hypertension, heart disease and diabetes, systolic blood pressure (SBP), diastolic blood pressure (DBP), the levels of triglycerides (TG), total cholesterol (Tcho), high-density lipoprotein cholesterol (HDL-C), low-density lipoprotein cholesterol (LDL-C), Aβ42 and P-tau181 in plasma. The overall cognitive status was assessed by neurologists with >5 years of working experience using the Mini-Mental State Examination (MMSE) and Montreal Cognitive Assessment (MoCA). CI was indicated by an MMSE score <27 or a MoCA score <26. Otherwise, a patients was considered to be in the NCI group (Nasreddine et al., 2005; Folstein, Folstein & McHugh, 1975).

MRI data were acquired using one of the five scanners used at the hospital during the study period. All scanning protocols included a T1-weighted gradient-echo sequence, fluid-attenuated inversion recovery (FLAIR) sequence, and T2-weighted sequence. The detailed scanning parameters are shown in Table S1.

WMH was defined as the areas of intensity signal on the T2-weighted FLAIR sequence (Wardlaw et al., 2013) divided into periventricular WMH (PVWMH) and deep WMH (DWMH) (Fazekas et al., 2002). PVWMH regions were defined as the regions within 1 cm from the ventricular border in each axial slice. DWMH was defined as the regions more than 1 cm from the edge of the lateral ventricle (Fig. 1A). Semiquantitative PVWMH and DWMH evaluations were performed by one senior radiologist. The evaluation was performed according to the Fazekas scale (score 0-3) (Fazekas et al., 2002) (Fig. S1). The patients were stratified into two different subgroups. The normal/mild subgroup included Fazekas grade 0-1, and the moderate/severe subgroup included Fazekas grade 2-3 (Peng et al., 2020).

Figure 1 (A) Distribution of deep white matter hyperintensities (DWMH) and periventricular white matter hyperintensities (PVWMH). (B) An example of the white matter hyperintensities (WMH) quantitative assessment in one of the normal cognitive Aβ- subject.

Quantitative WMH: The white matter lesions on T2 FLAIR images were segmented using a free and automated method (the lesion prediction algorithm (LPA)), which provided by the Lesion Segmentation Toolbox (LST) (version 2.0.15, www.statistical-modeling.de/lst.html) in SPM (SPM12, MATLAB v.2017b; MathWorks, Natick, MA). Based on this algorithms, a lesion probability score for each voxel was calculated (Weeda et al., 2019). A total of 0.01 cm3 was selected to be minimum extent threshold. The outputs of LPA were the lesion probability maps, which were binarized by using a threshold of 0.5, then lesion masks were generated. However, there were some error in the lesion masks generated by the algorithm, it was necessary to visually confirm and correct the false-negative or false-positive. The probability graph marked with errors was evaluated by two radiologists and imported into ITK-SANP software (Version 3.8.0; http://www.itksnap.org/pmwiki/pmwiki.php) for manual adjustment. WMH volume in cm3 was defined as the voxel size multiplied by the total number of voxels labeled as lesions in the cerebrum (Fig. 1B).

The traditional linear measurement method has been widely used for brain atrophy evaluation in the clinic and research (Jeong et al., 2016; Pontillo et al., 2020; Wu, 1985. The measurement was performed by two neuroradiologists. Cerebral atrophy includes central and cortical atrophy. The lateral ventricle body index (LVBI) was used to evaluate central atrophy. LVBI = b/a, a = minimum distance of lateral wall of bilateral lateral ventricle, and b = brain transverse diameter at the same level. Axial views of the brain T1WI were reviewed to locate and measure a and b (Fig. 2A), and the average value was taken. Central atrophy was assigned scores of 0-3 based on LVBI (Table S2). Cortical atrophy was considered present whenever two or more sulci each had a width >3 mm. Two or more sulci with the widest relative width of each lobe were selected, and the width was measured perpendicular to each sulci on the T1WI. Cortical atrophy was also scored on a scale ranging from 0-3 based on the maximum sulci width. Two observers measured the sulci widths of the frontal, parietal, temporal, and occipital lobes and assigned scores directly. In cases of disagreement, a final result will be determined after another measurement (Fig. 2B, Table S2). Observers scored and measured the neuroimaging markers in a blinded manner.

Figure 2 Linear measurement method of brain atrophy on axial T1WI. The minimum distance of lateral wall of the bilateral lateral ventricle (a) and transverse brain diameter at the same level (b) were measured (A), then lateral ventricle body index (LVBI) was obtained. In this case, LVBI = 2.95. Sulci width (B).

Statistical analysis

Participantcs were described into the NCI and CI groups. The test kit instructions and defined reference range of Aβ42 and p-tau-181 (Anqun Biological Engineering Co., Shenzhen, China) were used. A cutoff of 110 pg/ml was used to classify the patients as amyloid-positive (Aβ+) or amyloid-negative (Aβ-). A cutoff of 30 pg/ml was used to classify the patients as p-tau181 positive (tau+) or negative (tau-). The NCI and CI groups were further divided into Aβ-, tau- and Aβ-/tau- subgroups.

Baseline clinical characteristics were compared between CI and NCI groups. First, the Shapiro-Wilk test and visual inspection of histograms were applied to check the normality of the continuity variables. If it was a normally distributed continuous variable, presented as mean ± SD and analyzed by the Student’s t test; otherwise, they were presented as median (range) and tested using the Mann‒Whitney U test. For categorical variables, they were described as n (%) and assessed using the chi-square test. To compare differences in plasma and neuroimaging markers between CI and NCI groups, we performed multivariable logistic regression, in which the variables with P-values <0.10 in univariable analysis were included and variables with collinearity were excluded. Diagnostic sensitivity, specificity and accuracy of biomarkers was assessed using the receiver operator characteristic (ROC) analysis and area under the ROC curve (AUROC). For the comparison of the neuroimaging markers between different subgroups, we used the same statistical methods. Two-sided P values < 0.05 were considered statistically significant. All statistical analyses were performed with SPSS 25.0 (IBM Corp., Armonk, NY, USA) and MedCalc 18.2.1 (MedCalc Software bvba, Ostend, Belgium).

Results

Demographic characteristics of all enrolled patients

We prospectively enrolled 184 patients in this study, including 54 (29.4%) in the CI group and 130 (70.7%) in the NCI group (Fig. 3). The mean age (SD) of all patients was 82.7 (±3.4) years. There were no significant differences between the two groups for age, systolic blood pressure, diastolic blood pressure, blood lipid levels and the percentage of sex, history of hypertension, diabetes, and cardiac diseases (Table 1).

Figure 3 Flowchart of patient inclusion.

Table 1 Baseline characteristics of all patients (n = 184).

Characteristics	NCI (n = 130)	CI (n = 54)	P	
Age (years)	82 (81, 85)	82 (80, 85)	0.197	
Female, n (%)	55 (42.3)	15 (27.8)	0.065	
Hypertension, n (%)	86 (66.2)	38 (70.4)	0.578	
SBP (mm Hg)	147 ± 25	151 ± 25	0.288	
DBP (mm Hg)	80 ± 13	82 ± 14	0.302	
Diabetes, n (%)	35 (26.9)	16 (29.6)	0.709	
Cardiac disease, n (%)	35 (26.9)	13 (24.1)	0.689	
Tchoa	4.11 ± 1.05	4.16 ± 1.04	0.805	
TGb	1.21 ± 0.62	1.16 ± 0.61	0.658	
HDLc	1.17 ± 0.33	1.17 ± 0.33	0.981	
LDLc	2.40 ± 0.90	2.47 ± 0.89	0.634	
Notes:

a means 17 patients missed the date.

b means 19 patients missed the date.

c means 21 patients missed the date.

Data are presented as mean ± SD or median (Q25, Q75) or n (%). Statistics was based on the Student’s t-test or Mann–Whitney U test or the chi-square test.

NCI, non-cognitive impairment; CI, cognitive impairment; SBP, systolic blood pressure; DBP, diastolic blood pressure; Tcho, Total cholesterol; TG, triglycerides; HDL, high-density lipoprotein; LDL, low-density lipoprotein.

Comparison of neuroimaging markers between the NCI and CI groups

The interclass correlation coefficient (ICC) results of linear indicators showed good agreement between the two radiologists, ranging from 0.996 to 1.000 (Table S3).

Univariate logistic regression analysis showed the frequency of moderate and severe PVWMH was significantly higher in the CI group than in the NCI group (P < 0.001). Multivariate analysis demonstrated that moderate and severe PVWMH were related to CI (OR 2.857, [1.365–5.983], P = 0.005). The LVBI was 3.77 and 4.46 in the CI and NCI groups, respectively (P < 0.001). Multivariate analysis showed that LVBI with an OR of 0.413 (95% CI [0.243–0.700], P < 0.001) and cortical atrophy with an OR of 1.304 (95% CI [1.079–1.575], P = 0.006) were two related factors for CI.

The WMH volume, DWMH, TWMH, total atrophy and TWMH and total atrophy showed no significant difference in multivariate analysis. The frequencies of Aβ42+, P-tau 181+, and Aβ42+/P-tau181+ were not significantly different between the CI and NCI groups (all P > 0.05) (Table 2). The AUROC was 0.782 by using the combined model (PVWMH, LVBI and CA) to distinguish between CI and NCI, with 68.5% sensitivity and 78.5% specificity (Tables 2 and S4, and Fig. 4A).

Table 2 Comparison of the neuroimaging markers between the two groups (n = 184).

Characteristics	Univariable analysis	Multivariable analysis	
NCI (n = 130)	CI (n = 54)	P	OR (95% CI)	P	
WMH volume (cm3)	19.15 (8.45, 35.00)	27.73 (11.69, 56.73)	0.001*	1.00 [0.985–1.016]	0.957	
WMH (Fazekas score)						
DWMH (2, 3)	48 (36.9)	32 (59.3)	0.005*	1.746 [0.562–5.428]	0.335	
PVWMH (2, 3)	58 (44.6)	38 (70.4)	0.001*	2.857 [1.365–5.983]	0.005*	
TWMH (3-6)	60 (46.2)	38 (70.4)	0.003*	-	-	
Brain atrophy						
LVBI	4.46 ± 0.94	3.77 ± 0.67	<0.001*	0.413 [0.243–0.700]	0.001*	
Cortical atrophy	2.88 ± 1.74	4.20 ± 2.16	<0.001*	1.304 [1.079–1.575]	0.006*	
Total atrophy	3.43 ± 2.08	5.44 ± 2.36	<0.001*	–	-	
TWMH and total atrophy	6.62 ± 2.53	9.33 ± 2.98	<0.001*	–	-	
Aβ1-42+2, n (%)	54 (41.53)	21 (33.89)	0.886			
P-tau 181+3, n (%)	37 (28.46)	14 (27.45)	0.892			
Aβ1-42+/P-tau+3, n (%)	26 (20.00)	13 (25.49)	0.419			
Notes:

* P < 0.05.

Data are presented as mean ± SD or median (Q25, Q75) or n (%). Statistics was based on the Student’s t-test or Mann–Whitney U test or the chi-square test. Variables with collinearity were excluded, and variables with P-values <0.10 in the univariable logistic regression analyses were included in the multivariable analysis.

NCI, non-cognitive impairment; CI, cognitive impairment; WMH, white matter hyperintensities; DWMH, deep white matter hyperintensity; PVWMH, periventricular white matter hyperintensity; TWMH, total white matter hyperintensity; LVBI, Lateral ventricle body index; OR, odds ratio. Aβ, amyloid beta; P-tau, phosphorylated tau protein. 2,3,3means the number of missing data.

Figure 4 Receiver operating characteristic (ROC) curve of MRI makers for differentiating cognitive impairment from non-cognitive impairment groups and for different subgroups.

(A) Receiver operating characteristic (ROC) curve of the combined PVWMH, LVBI, and cortical atrophy for differentiating CI from NCI groups (n = 184). (B) Receiver operating characteristic (ROC) curve of the combined PVWMH and LVBI for the Aβ- subgroups (n = 107). (C) Receiver operating characteristic (ROC) curve of the combined PVWMH and LVBI for the tau- subgroups (n = 130). (D) Receiver operating characteristic (ROC) curve of LVBI for the Aβ-/tau-subgroups (n = 94).

Comparison of neuroimaging markers between NCI and CI in the A β- and tau- subgroups

A total of 58.8% (107/182) and 71.82% (130/181) of patients were Aβ- and tau-, respectively. In the Aβ- subgroup (n = 107), the LVBI was 3.86 and 4.45 in the CI and NCI groups, respectively (P = 0.002). Multivariate analysis showed that LVBI was related to CI (OR = 0.484, 95% CI [0.258–0.907], P = 0.024). The proportion of moderate and severe PVWMH had a certain effect on CI, with an OR of 2.632 (95% CI [1.033–6.706], P = 0.043) after multivariate analysis, although the difference was no significant in univariate analysis (P = 0.089). The AUROC was 0.734 by combining LVBI and PVWMH to differentiate CI and NCI in the Aβ- subgroup, the sensitivity and specificity were 71.0% and77.6%, respectively (Tables 3 and S4, and Fig. 4B).

Table 3 Comparison of neuroimaging markers between the Aβ- subgroups (n = 107).

Characteristics	Univariable analysis	Multivariable analysis	
NCI (n = 76)	CI (n = 31)	P	OR	P	
WMH volume (cm3)	20.24 (8.07, 34.54)	19.55 (10.02, 49.92)	0.454	–	–	
WMH (Fazekas score)						
DWMH (2,3)	28 (36.8)	17 (54.8)	0.087	1.184 (0.323, 4.334)	0.799	
PVWMH (2,3)	33 (43.4)	8 (25.8)	0.089	2.632 (1.033, 6.706)	0.043*	
TWMH (3-6)	35 (46.1)	21 (67.7)	0.042*	-	-	
Brain atrophy						
LVBI	4.45 ± 0.88	3.86 ± 0.78	0.002*	0.484 (0.258, 0.907)	0.024*	
Cortical atrophy	2.97 ± 1.81	3.81 ± 2.07	0.041*	1.172 (0.922, 1.490)	0.196	
Total atrophy	3.51 ± 2.17	4.94 ± 2.39	0.004*	–	–	
TWMH and total atrophy	6.67 ± 2.53	8.65 ± 3.09	0.001*	–	-	
Notes:

* P < 0.05.

Data are presented as mean ± SD or median (Q25, Q75) or n (%). Statistics was based on the Student’s t-test or Mann–Whitney U test or the chi-square test. Variables with collinearity were excluded, and variables with P-values <0.10 in the univariable logistic regression analyses were included in the multivariable analysis.

NCI, non-cognitive impairment; CI, cognitive impairment; WMH, white matter hyperintensities; DWMH, deep white matter hyperintensity; PVWMH, periventricular white matter hyperintensity; TWMH, total white matter hyperintensity; LVBI, Lateral ventricle body index; Aβ-, amyloid beta negative; OR, odds ratio.

In the tau- subgroup (n = 130), univariate analysis showed that the percentage of moderate and severe PVWMH was significant different in CI and NCI groups (70.3% vs 46.2%, respectively, P = 0.013). Moderate and severe PVWMH was a related factor for CI in multivariate analysis (OR 2.816, [1.180–6.717], P = 0.020). For patients in the CI and NCI groups, the LVBI was 3.90 and 4.49, respectively (P = 0.001). After multivariate analysis, the LVBI was demonstrated to be a relevant factor for CI with an OR of 0.496 (95% CI [0.279–0.881], P = 0.017). To distinguish between CI and NCI in the tau- subgroup by the model combining PVWMH and LVBI, the AUROC was 0.757, the sensitivity and specificity were 83.8% and 65.6%, respectively (Tables 4 and S4, and Fig. 4C).

Table 4 Comparison of neuroimaging markers between tau- subgroup (n = 130).

Characteristics	Univariable analysis	Multivariable analysis	
NCI (n = 93)	CI (n = 37)	P	OR	P	
WMH volume (cm3)	20.24 (7.63, 34.72)	22.15 (11.48, 47.83)	0.187	–	–	
WMH (Fazekas score)						
DWMH (2,3)	37 (39.8)	20 (54.1)	0.139	–	–	
PVWMH (2,3)	43 (46.2)	26 (70.3)	0.013*	2.816 (1.180, 6.717)	0.020*	
TWMH (3-6)	45 (48.4)	26 (70.3)	0.024*	0.480 (0.156, 1.478)	0.201	
Brain atrophy						
LVBI	4.49 ± 0.92	3.90 ± 0.72	0.001*	0.496 (0.279, 0.881)	0.017*	
Cortical atrophy	2.62 ± 1.79	3.73 ± 2.05	0.003*	1.247 (0.998, 1.559)	0.052	
Total atrophy	3.17 ± 2.19	4.78 ± 2.32	<0.001*	–	–	
TWMH and total atrophy	6.42 ± 2.55	8.35 ± 2.71	<0.001*	–	-	
Notes:

* P < 0.05.

Data are presented as mean ± SD or median (Q25, Q75) or n (%). Statistics was based on the Student’s t-test or Mann–Whitney U test or the chi-square test. Variables with collinearity were excluded, and variables with P-values <0.10 in the univariable logistic regression analyses were included in the multivariable analysis.

NCI, non-cognitive impairment; CI, cognitive impairment; WMH, white matter hyperintensities; DWMH, deep white matter hyperintensity; PVWMH, periventricular white matter hyperintensity; TWMH, total white matter hyperintensity; LVBI, Lateral ventricle body index; tau-, phosphorylated tau protein negative; OR, odds ratio.

Comparison of neuroimaging markers between NCI and CI in the A β-/tau- subgroup

Ninety-four (51.09%) patients were both Aβ negative and tau negative. The LVBI was 3.89 ± 0.79 and 4.46 ± 0.90 in CI and NCI, respectively (P = 0.005). Multivariate analysis showed that the LVBI was significantly different, with an OR of 0.442 (95% CI [0.239–0.819], P = 0.009). Using LVBI to discriminate CI from NCI in the Aβ-/tau- subgroup, the AUROC was 0.720, the sensitivity and specificity were 69.0% and 76.9%, respectively (Tables 5 and S4, and Fig. 4D).

Table 5 Comparison of between the Aβ- and tau- subgroups (n = 94).

Characteristics	Univariable analysis	Multivariable analysis	
NCI (n = 65)	CI (n = 29)	P	OR	P	
WMH volume (cm3)	20.18 (7.63, 33.99)	16.60 (9.76, 34.04)	0.650	–	–	
WMH (Fazekas score)						
DWMH (2,3)	31 (47.7)	8 (27.6)	0.068	1.241 (0.338, 4.551)	0.745	
PVWMH (2,3)	37 (56.9)	11 (37.9)	0.089	2.245 (0.871, 5.786)	0.094	
TWMH (3-6)	39 (60.0)	11 (37.9)	0.048*	–	–	
Brain atrophy						
LVBI	4.46 ± 0.90	3.89 ± 0.79	0.005*	0.442 (0.239, 0.819)	0.009*	
Cortical atrophy	2.89 ± 1.84	3.62 ± 2.01	0.088	1.145 (0.893, 1.468)	0.287	
Total atrophy	4.06 ± 2.33	3.31 ± 2.21	0.146	–	–	
TWMH and total atrophy	7.52 ± 2.81	6.34 ± 2.41	0.053	–	–	
Notes:

* P < 0.05.

Data are presented as mean ± SD or median (Q25, Q75) or n (%). Statistics was based on the Student’s t-test or Mann–Whitney U test or the chi-square test. Variables with collinearity were excluded, and variables with P-values <0.10 in the univariable logistic regression analyses were included in the multivariable analysis.

NCI, non-cognitive impairment; CI, cognitive impairment; WMH, white matter hyperintensities; DWMH, deep white matter hyperintensity; PVWMH, periventricular white matter hyperintensity; TWMH, total white matter hyperintensity; LVBI, Lateral ventricle body index; Aβ-/tau-, both amyloid-beta and phosphorylated tau protein negative; OR, odds ratio.

Discussion

The results suggest that conventional structural MRI parameters, such as WMH, LVBI and cortical atrophy, are relevant factors for cognitive impairment, while plasma Aβ42 and P-tau181 are not related to cognitive impairment. PVWMH, LVBI, and cortical atrophy may be used as indicators of cognitive impairment in elderly patients (≥75 years).

Our study included only older patients (≥75 years), in whom the presence of WMH and brain atrophy would be more common and relevant. A higher WMH burden is related to many factors, such as age, hypertension, and diabetes (Guevarra et al., 2020; Yuan et al., 2021). In our study, those characteristics showed no difference between the NCI and CI groups. Nevertheless, the PVWMH burden was higher in CI patients than in NCI patients. The percentages of PVWMHs with Fazekas scores of 2-3 were 44.6% and 70.4% in the NCI and CI groups, respectively. These results are supported by the literature (Guevarra et al., 2020; Yuan et al., 2021; Kloppenborg et al., 2014). Guevarra et al. (2020) showed that total WMH alone was associated with poorer MoCA scores in ≥70-year-old patients. Tubi et al. (2020) found that larger total WMH volumes and PVWMH were significantly related to a worse diagnosis, but not DWMH was significantly associated with the diagnosis. Griffanti et al. (2018) showed that PVWMH was related to global cognition. However, there are also some differences among these earlier findings. We found that the burden of TWMH or DWMH was not significantly different between CI and NCI group in multivariable analysis. A possible explanation for the differences among these studies might be that the patients with CSVD in the present study and their age were older than those in other studies. WMH is associated with age, DWMH and PVWMH are related to severe myelin loss and increased microglia activity (Simpson et al., 2007), which perhaps had a greater impact on PVWMH than DWMH in CSVD patients. Some studies have indicated that DWMH and PVWMH might also have different etiologies: DWMH might be related to axonal loss, arteriolosclerosis, and body mass index, while PVWMH might be associated with plasma leakage, arterial pressure,decline in total cerebral blood flow, blood‒brain and barrier permeability, which may be all causes of CSVD (Griffanti et al., 2018; Simpson et al., 2007; Haller et al., 2013). Another explanation might be the different evaluation methods for WMH (Guevarra et al., 2020).

Previous studies have shown that brain atrophy can occur in healthy older or cognitive impairment patients (Camarda et al., 2018; Aljondi et al., 2019). In this study, individuals with cognitive impairment had more severe atrophy (including central atrophy and cortical atrophy) than those in the NCI group. The LVBI was 3.77 and 4.46, the mean scores of cortical atrophy were 4.20 and 2.88 in the CI and NCI groups, respectively. The diagnostic performance of combined WMH and brain atrophy was higher than that of WMH and brain atrophy alone. The cognitive status of the elderly over 75 years old was taken as the endpoint event, and it was found that WMH and brain atrophy were related to cognitive impairment in the elderly at this stage. Previous studies (Guevarra et al., 2020; Aljondi et al., 2019) in people under 75 years of age have also found that WMH and brain atrophy were associated with cognitive impairment. Therefore, we believe that WMH and brain atrophy are useful biomarkers for evaluating or screening cognitive function, and their early assessment and dynamic observation may have important clinical significance.

In primary care, validated biomarkers can determine which patients should undergo more specialized diagnostic testing, thereby saving time and money. Blood-based biomarkers have the potential to be cost-effective, noninvasive, and easily performed, since blood sampling has became a routine clinical examination process. Early studies validated that plasma Aβ42 and p-tau181 are related to multiple Alzheimer’s disease-associated cognitive domains and AD-related CSF biomarkers at all clinical stages of AD (Kapasi, DeCarli & Schneider, 2017; Palmqvist et al., 2019; Qu et al., 2023). Simrén et al. (2021) found that higher plasma P-tau181 was related to cognitive decline in patients with CSVD. Qu et al. (2023) indicated that Aβ42 in plasma levels were correlated with CSVD and memory domains in patients with an average age of 67.5 years. They used a new sandwich ELISA to detect plasma Aβ42 and oligomeric Aβ42. However, some studies examining CSF or plasma have suggested no difference in plasma Aβ42 and p-tau181 levels between AD and controls (Luo et al., 2018; Hansson et al., 2012). Consistently, in our study, there were no significant differences in the percentages of plasma Aβ42, P-tau181, and tau/Aβ42 between the NCI and CI groups. This inconsistency might be mainly due to the differences in age, disease of the patients, detection method and indicators of CI. In our study, plasma Aβ42 and P-tau181 might be nonspecific markers for neurodegeneration in people ≥75 years of age.

Previous studies showed that WMH volume could predict amyloid positivity in NCI and CI individuals (Kandel et al., 2016; Palhaugen et al., 2021). In the present study, 58.8% (107/182) of the patients were Aβ-, and 71.8% (130/181) were tau-. As an exploratory analysis, neuroimaging markers were examined in the Aβ-, tau-, and Aβ-/tau- subgroups. WMH loads were not significantly different between the NCI and CI individuals in the Aβ-/tau- subgroups. PVWMH and central atrophy were most severe in the CI-Aβ- and CI-tau- subgroups. These results suggest that neuroimaging markers (especially atrophy) might be more useful for patients ≥75 years with normal levels of plasma Aβ and p-tau181, which is consistent with earlier literature (Deters et al., 2017; Hansson et al., 2021).

There are several limitations in our study. First, the patients were from a single center, with a small sample size. Second, the cross-sectional nature of the data limits the scope of interpretation of the findings. A longitudinal study aimed at the dynamics of neuroimaging markers and the levels of plasma amyloid and tau will help us clarify the relationship between these factors and cognitive impairment. Third, only patients aged ≥75 years were included in this study. Cognitive impairment often coexists with cerebrovascular disease in elderly individuals, and many patients were excluded, such as those with massive brain infarction. Finally, because of the small sample size, there wasn’t detailed grouping of cognitive impairment, such as grouping according to the different stages of cognitive impairment or according to functions (including memory, attention, executive ability, speech function, visual spatial structure function and clinical daily living ability) impaired and the degree of impairment. In future studies, we would like to expand the sample size or carry out a multicenter study, apply more advanced MRI technology/parameters, and conduct longitudinal cohort studies on patients with different stages or types of cognitive impairment, as well as studies on normal populations under 75 years old.

Conclusions

Among older adults (≥75 years), plasma Aβ42 and p-tau181 were not associated with cognitive impairment, while MRI parameters, including PVWMH, LVBI and cortical atrophy, were related to cognitive impairment. In this study, the cognitive statuses of the elderly over 75 years old were taken as the endpoint event. It might have more important clinical significance for early assessment and dynamic observation of these routine MRI markers, but future studies are needed to verify this hypothesis.

Supplemental Information

Supplemental Information 1 Raw data.

Click here for additional data file.

Supplemental Information 2 Supplementary Tables.

Click here for additional data file.

Supplemental Information 3 Representations of grades 0, 1, 2, and 3 for DWMH and PVWMH.

Click here for additional data file.

Supplemental Information 4 To investigate the clinical application value of plasma amyloid Aβ1-42 (Aβ42), phosphorylated tau 181 (p-tau181) and conventional MRI markers for cognitive impairment (CI) in patients aged ≥75 years, we conducted a retrospective study.

Patients with a clinical diagnosis of CI (n = 54) (yellow) and without CI (n = 130) (blue) were enrolled, clinical variables including plasma indicators (Aβ42 and p-tau181) and conventional structural MRI parameters were collected and analyzed. We found that the CI group (yellow) had higher WMH load, smaller LVBI value, and more severe cortical atrophy compared to the non-CI (NCI) group (blue), while plasma Aβ42 and p-tau181 showed no significant difference between the two groups. The combined model including PVWMH, LVBI, and cortical atrophy to detect CI and NCI showed an area under the ROC curve is 0.782.

Click here for additional data file.

Additional Information and Declarations

Competing Interests

Author Contributions

Human Ethics

Data Availability

Yaqiong Ge is employed by GE Healthcare. The authors declare that they have no competing interests.

Wei Wang conceived and designed the experiments, performed the experiments, analyzed the data, prepared figures and/or tables, authored or reviewed drafts of the article, and approved the final draft.

Lin Shi performed the experiments, analyzed the data, prepared figures and/or tables, and approved the final draft.

Hong Ma performed the experiments, analyzed the data, authored or reviewed drafts of the article, and approved the final draft.

Shiguang Zhu performed the experiments, analyzed the data, authored or reviewed drafts of the article, and approved the final draft.

Yaqiong Ge performed the experiments, analyzed the data, authored or reviewed drafts of the article, and approved the final draft.

Kai Xu conceived and designed the experiments, performed the experiments, authored or reviewed drafts of the article, and approved the final draft.

The following information was supplied relating to ethical approvals (i.e., approving body and any reference numbers):

This retrospective analysis was approved by the ethics committee of the Affiliated Hospital of Xuzhou Medical University (XYFY2022-KL331-01). The requirement for informed consent was waived by the committee.

The following information was supplied regarding data availability:

The raw data are available in the Supplemental Files.

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
