# Peer review of "Comparison of the clinical value of MRI and plasma markers for cognitive impairment in patients aged ≥75 years: a retrospective study"

_PeerJ, doi:10.7717/peerj.15581_

## Round 0.1 · original submission · Major Revisions

A detailed point-to-point response letter (editor + reviewers), and manuscript with tracked font are necessary for further process when they submit their revised version of paper. Please upload the response letter as a PDF file in the Supplementary materials in the online submission system. Please give detailed response. Please write down what changes have been made in the response letter, rather than reading the manuscript to find what has been revised. And please note that any uncompleted or improper corrections by the authors during this revision may lead to rejection.

1. The language needs revision by a fluent speaker, accompanied with a certificate of language editing service and a manuscript with tracked editing records as a Suppl.
2. Please use https://www.figcheck.com/imagecheck or similar free/paid picture duplicate checking software to detect all figures and upload the examining PDF report as a Supplement file.
3. Similar studies were found in PubMed. Authors shall cite part of them and discuss what is new and different from this article.
4. State n = ? in Section Statistical analysis and all figure legends.
5. Please provide a duplicate check report by authors as a supplementary file (total < 20%, each < 2%).

Reviewer 1 ·

Basic reporting

a. In the result, Lines 166-167: “This study included 203 patients, but 19 were excluded; therefore, 184 patients were included finally, including ……”. Since the authors have showed a participant enrollment flowchart in Figure 3, I think if the authors only show the number of patients final enrolled in the study, it will be more clearly.
b. In the result of abstract and the manuscript (Lines 172-175) , “Univariable logistic regression analysis showed that Aβ42+, P-tau 181+, and Aβ42+ and P-tau181+ were not significantly different between the cognitive impairment and NCI groups (all P>0.05)”. What is the meaning of this sentence? Did you want to show that there is no significant difference in the frequency of biomarkers positive patient between the cognitive impairment and NCI groups ? please confirm it.
c. "Among individuals ≥75 years, plasma Aβ42 and P-tau181 might be associated with cognitive impairment, while MRI parameters, such as WMH, LVBI, and brain atrophy, may be potential indicators to predict cognitive impairment." Based on the abstract results (plasma biomarkers were not significantly associated with cognition) this sentence is contradictory when it says "plasma Aβ42 and P-tau181 might be associated with cognitive impairment". Please explain your conclusion.
d. The work is based on the fact that "An affordable, non-invasive means is essential for clinical application and research, especially for large-scale screening programs and longitudinal studies." and their aim is to "investigate the predictive value of plasma Aβ1-42 and p-tau181 and conventional structural MRI parameters for cognitive impairment in patients aged >75 years." Since the cost of MRI examination is higher than that of CT, I do not think MRI is an affordable mean? How about the plasma biomakers? Please discuss this point.
e. Is this CI (Line 182) an abbreviation of cognitive impairment ? It seems to appear for the first time. Is there a description in the previous article?
f. In the figure 4, what is the meaning of “mode1/2/3/4”? Is its meaning different groups? Please explain it or correct the figure.

Experimental design

The authors retrospectively identify 54 patients over 75 years of age with cognitive impairment and 130 patients without cognitive impairment to evaluated the clinical value of plasma Aβ1-42, p-tau181, and MRI parameters for cognitive impairment. By testing plasma biomarkers (Aβ1-42 and p-tau181) and conventional structural MRI markers (WMH and brain atrophy), WMH, LVBI and complete atrophy were identified as three imaging specific markers through multivariate regression analysis. The combined diagnostic value of MRI parameters in the three subgroups of Aβ (-), p-tau181 (-) and Aβ (-) +p-tau181 (-) was analyzed by fitting ROC curve, and the specific diagnostic markers for cognitive impairment patients over 75 years old were explored.
a. The main outcome of the analysis is cognition, but the criteria used is not clearly stated. Which cut-off was used for MMSE and MoCA to define CI?
b. The method used for the quantification of plasma biomarkers was not described in the methods, as well as the definition of its cutoff points.
c. In this study, the authors evaluated brain atrophy by using T1WI images, rather than T2WI or FLAIR images. Please give explanation for this.

Validity of the findings

a. In the Conclusions, Lines 258-259: “ while plasma Aβ42 and p-tau181 were not associated with cognitive impairment” is contradictory to the conclusion in the abstract.
b. In the Results, Lines 173-175: “while Aβ42+, P-tau 181+, and Aβ42+ and P-tau181+ had not significant difference between cognitive impairment and NCI groups (all P>0.05)”. What does AB+ and tau+ mean? Is it the Aβ and tau statuses? What is the "Aβ42+ and P-tau181+" group? Since there are two “and” here, it is confused.

Additional comments

This study is interesting and I think it is of clinical significance.
a. In view of the limitations of this study, what solutions do authors have?
b. Be careful with language about “predictive”,perhaps saying that they are associated is more reasonable since this is a cross-sectional analysis.
c. In this study, patients ages ≥75 years were enrolled. However, in the abstract (background part) and introduction( line 84 ), the authors should correct “>75 years” to “≥75 years”. Please also unify the use of “Aβ42 and Aβ1-42” throughout the manuscript.
d. Please revise the punctuation error in line 64 “WMH is observed in a relatively large proportion of older adults (Wardlaw et al., 2013). and…”
e. In the last paragraph of “Introduction”, the authors should introduce their study a little more, not only the purpose, but also the conclusion and significance of their study.
f. In the last paragraph of “Discussion”: “Finally, there was no detailed grouping of cognitive impairment because of the small sample size.” Did the authors mean the sample size is too small to be grouped by other methods. What are the other grouping methods?

Reviewer 2 ·

Basic reporting

The article was written in very clear and understandable English, making it accessible to a wide range of readers. However, the consent forms sent by the patients were in Chinese, which could make it difficult for non-Chinese speakers to understand and assess the patients' level of informed consent.

In terms of the article structure, while the introduction and material sections were well-written and informative, the result section was disappointingly brief. The authors failed to provide a clear explanation of their findings, which can be confusing for readers trying to understand the study's implications. This lack of clarity can be attributed to the insufficient detail provided on the methodology, data processing, and statistical analysis used to arrive at the study's conclusions.

The article would have benefited from a more robust and detailed discussion of the results section. This could have involved a more detailed explanation of the statistical tests used, the significance of the findings, and how they fit within the existing body of research on the topic. Such explanations would help readers understand the study's contribution to the field and how it advances our understanding of the research question at hand.

In conclusion, while the report was written in accessible English and contained useful information, there is room for improvement in terms of the clarity of the results section and the structure of the article as a whole. By providing more detailed explanations of the methodology, data used and data analysis, and statistical tests, the authors could have improved the article's usefulness to readers.

Experimental design

The experimental design mentions using a DWI sequence but there wasn't anything mentioned about the result of using that sequence in the paper itself.

Validity of the findings

The article discusses the role of cerebral small vessel disease as a primary contributor to vascular cognitive impairment (VCI) and the use of neuroimaging markers as common imaging markers in detecting this condition. Although Alzheimer's is responsible for more cases of Cognitive impairment in older adults, this study focuses on cerebral small vessel disease patients.

The article mentions two common neuroimaging markers used to detect VCI, but there is a lack of clarity on how these biomarkers are independent risk factors for cognitive impairment. Despite many studies showing that neuronal death is the root cause of both white matter hyperintensities (WMH) and cognitive impairment, the basis for concluding that these biomarkers are independent risk factors is not apparent in the paper.

The observational measurement of sulci widths is not an objective and quantitative method for assessing brain atrophy. It should be noted that software tools such as Freesurfer and ANTS can be used to process these measurements. However, the result section would benefit from more detailed data processing and specifically addressing how the sulci widths measurements were used to support the study's findings. (lines 141-142)

In line 219, the article mentions that cognitive impairment adults have more severe brain atrophy than the NCI group, but there is no numerical information provided to indicate how much more severe the atrophy is. Overall, the article provides some insights into the role of cerebral small vessel disease and neuroimaging markers in VCI, but there is a need for more clarity on the knowledge gap they addressed, the basis for conclusions, and the study's contribution to the field

Reviewer 3 ·

Basic reporting

In the present study, Wang et al., show that White Matter Hyperintensities (WMH) and Brain atrophy are effective biomarkers to study cognitive decline in patients >75 years, whereas, well studied biomarkers such as Ab1-42 and tau are not as effective as biomarkers. I would commend the authors for a detailed and comprehensive manuscript, well reported arguments and their supplementary information.

The article was well written, but enough references, however, I would suggest providing references for some cutoffs (such as for line 148, 149 for threshholding tau and Abeta), and where these cutoffs came from, since it forms the premise of the analysis.

Experimental design

The experiments design and methods is well written. Please note the title says "plasma makers" which should be plasma "markers".

Validity of the findings

1) In line 222, authors mention that these two markers are might be useful for markers for predicting changes earlier in the life, however, this should be validated by choosing patients <75 years. Major changes during >75 years might have a causal impact on brain atrophy and WMH, rather than the other way around.

2) For the predictive power, I want to ask whether the authors imply causality for cognitive impairments due to these changes, and if there are more references to support these changes early on during midlife than late life changes.

I believe the manuscript is well written.

---

## Round 0.2 · Minor Revisions

A detailed point to point response letter (editor + reviewers), manuscript with tracked font are necessary for further process when they submit their revised version of paper. Please upload the response letter as a PDF file in the Supplementary materials in the online submission system. Please give detailed response. Please write down what changes have been made in the response letter, rather than reading the manuscript to find what has been revised. And please note that any uncompleted or improper corrections by the authors during this revision may lead to rejection.
1. A graphic abstract is needed to add as a main figure.
2. Must use institution emails for corresponding authors in the title page, as the corresponding author is in a university-affiliated hospital.

Reviewer 1 ·

Basic reporting

no comment

Experimental design

no comment

Validity of the findings

no comment

Additional comments

The manuscript has been well improved, and I think my comments have been addressed.

Reviewer 2 ·

Basic reporting

I'm okay with the language, it is clear. It seems that they have resolved most problems of their result section, but the conclusion still needs more expansion and explanation of the work.

Experimental design

no comment

Validity of the findings

I'm happy with their response and the edits they have made

---

## Round 0.3 · accepted · Accept

The authors have addressed all questions.